# Increasing the Transport of Celecoxib over a Simulated Intestine Cell Membrane Model Using Mesoporous Magnesium Carbonate

**DOI:** 10.3390/molecules26216353

**Published:** 2021-10-21

**Authors:** Johan Gómez de la Torre, Christel Bergström, Teresa Zardán Gómez de la Torre

**Affiliations:** 1Division of Nanotechnology and Functional Materials, Department of Materials Science and Engineering, Uppsala University, P.O. Box 534, SE-751 21 Uppsala, Sweden; johan.gomezdelatorre@gmail.com; 2Drug Delivery, Department of Pharmacy, Uppsala University, P.O. Box 580, SE-751 23 Uppsala, Sweden; christel.bergstrom@farmaci.uu.se

**Keywords:** mesoporous materials, magnesium carbonate, poorly soluble drugs, celecoxib, Caco-2 cell membrane, drug release

## Abstract

In the current work, mesoporous magnesium carbonate (MMC) was used to suppress crystallization of the poorly soluble drug celecoxib (CXB). This resulted in both a higher dissolution rate and supersaturation of the substance in vitro as well as an increased transfer of CXB over a Caco-2 cell membrane mimicking the membrane in the small intestine. The CXB flux over the cell membrane showed a linear behavior over the explored time period. These results indicate that MMC may be helpful in increasing the bioavailability and obtaining a continuous release of CXB, and similar substances, in vivo. Neusilin US2 was used as a reference material and showed a more rapid initial release with subsequent crystallization of the incorporated CXB in the release media. The presented results form the foundation of future development of MMC as a potential carrier for poorly soluble drugs.

## 1. Introduction

Oral administration of drugs is the preferred and most frequently used route for administration since it is non-invasive and allows patients to easily manage the medication outside of hospitals or healthcare centers. The oral bioavailability of an active pharmaceutical ingredient (API) depends mainly on its aqueous solubility and permeability across the intestinal barrier. Many APIs suffer from poor aqueous solubility, which limits their uptake in the gastrointestinal tract and, therefore, their therapeutic effect when taken orally [1]. The problem is significant: about 40% of newly marketed drugs are poorly soluble, and 80–90% of drug candidates in the R&D pipeline fail because of solubility problems [2,3,4]. Numerous formulation strategies have been developed, such as salt formulations, API particle reduction, use of solubilizers, and lipid-based formulations [5,6]. The choice of method depends on the physiochemical properties of the specific API, and there is still a need for more alternatives in the formulation toolbox to make products out of promising drug candidates.

Nanotechnological advances provide new possibilities in the medical field, and this has started to draw attention to the area of drug development because of their ability to increase the apparent solubility of poorly soluble drugs. Various nanomaterials are being investigated as potential drug carriers, and one application is to use mesoporous materials to formulate APIs in their amorphous states [7,8,9,10,11,12]. Generally, amorphous APIs have higher apparent solubilities than those of their crystalline counterparts and can therefore provide a desired therapeutic effect where the crystalline form cannot. However, because of their metastable nature, amorphous APIs are driven to recrystallize to the more energetically favorable crystal form if not stabilized in the formulation. When incorporated in a mesoporous structure, recrystallization of the amorphous API is suppressed [12,13].

Mesoporous magnesium carbonate (MMC) is an X-ray amorphous material that can be synthesized without the use of any surfactants as pore templating agents [14]. It has a large surface area and pore volume, and the pores can be tuned during the synthesis between 2 and 20 nm [14,15,16]. In addition, the material is biocompatible [17]. MMC has also been shown to be able to stabilize several APIs in their amorphous states, resulting in enhanced apparent solubility and dissolution rate of APIs when performing in vitro dissolution tests [13,16,18,19].

The objective of this study is to investigate how the previously demonstrated improvement in dissolution kinetics of the poorly soluble model substance celecoxib (CXB) when formulated with MMC affects the flux of CXB in a Caco-2 cell model. The Caco-2 cell model is a well-characterized intestinal in vitro model that make it possible to evaluate the ability of APIs to cross the intestinal barrier. This cell model is widely used in the pharmaceutical industry during drug discovery and development as a predictive tool for the oral absorption of API candidates. [20,21,22,23]. A flux increase over the Caco-2 cells could potentially indicate that the bioavailability of CXB can be improved in vivo.

In this study, the in vitro release, dissolution and transfer over a Caco-2 cell membrane of CXB formulated with MMC are studied and compared to the behavior of crystalline (free) CXB and CXB formulated with Neusilin US2, a synthetic, amorphous magnesium aluminometasilicate. Neusilin US2 is a commercially available excipient listed in the US Pharmacopeia [24]. Due to its porous nature, large surface area and commercial availability, Neusilin US 2 is a suitable reference material.

## 2. Results and Discussion

### 2.1. Material Characterization

The pore size distributions for the studied materials are presented in Figure 1. The surface area and the pore volume are also displayed in the figures. The surface area was calculated using the multipoint Brunauer–Emmett–Teller (BET) method. [25] The unloaded samples are highly porous with a BET surface area of 332 and 436 m^2^/g for MMC and Neusilin, respectively. The pore size distribution of MMC shows a large peak centered at around 6 nm and a small peak that corresponds to pores of about 1.4 nm in diameter. The Neusilin sample has a broader pore size distribution where the peak is centered in between 10 and 20 nm. A reduction in BET surface area and pore volume for the drug-loaded samples were observed, which indicates that CXB was successfully loaded into both MMC and Neusilin. MMC-CXB still remained highly porous, which is an indication that CXB was loaded inside the porous structure rather than on the surface. If CXB was loaded on the surface of the material, it would have a greater negative impact on the surface area.

Figure 2 shows X-ray powder diffraction (XRD) patterns for the all studied materials, where Figure 2a shows free and loaded CXB, Figure 2b shows the MMC samples, and Figure 2c shows the Neusilin samples.

The peaks observed in Figure 2b correspond to unreacted MgO in the MMC material as observed in earlier studies [14,15]. What seems to be a higher background for the filled MMC sample compared to the empty sample at around 20°–30° indicates the presence of amorphous CXB. The arrows in Figure 2c indicate peaks corresponding to traces of crystalline CXB in the loaded Neusilin sample. The absence of clear peaks for CXB in both loaded samples indicates that CXB is mainly amorphous in both samples. This supports the results from N_2_ adsorption measurements, showing a reduction in pore volume after loading, which indicates that the CXB had actually entered the MMC and Neusilin pore structure.

Differential scanning calorimetry (DSC) results confirm the amorphous state of CXB in the loaded samples (see Figure 3). Crystalline traces of CXB can be seen in both loaded samples as small endothermic events at 164 °C that correspond to the melting point of CXB. The degree of crystalline CXB present in MMC and Neusilin was calculated to be 0.4 and 2%, respectively. This was calculated by comparing the melting enthalpy for crystalline and loaded CXB.

In order to confirm the CXB loading degree in the CXB-loaded samples, thermal gravimetric calorimetry (TGA) was conducted on pure CXB and unloaded and loaded samples.

The weight loss of the loaded carriers was compared with the weight loss of the empty carrier materials and pure CXB (see Figure 4). From the figure, it is clear that Neusilin and MMC lost 17 and 43% of their respective weights when heated up to 800 °C, while CXB lost 66% of its weight at around 350–400 °C. From this, the theoretical weight loss of the filled samples was calculated using Equation (1) as shown in Section 3.5. For the calculations, it was assumed that the theoretical drug loading degree is 33.3 wt% (as described in Section 3.3). The calculated weight losses for the MMC-CXB and N-CXB samples are 50 and 33%, respectively. From the TGA data, it is clear that MMC-CXB and N-CXB samples lost 48 and 33% of their weight, respectively, which is in almost complete agreement with the theoretical results. This indicates that the drug loading degree of the two formulations is approximately 33 wt%. Hence, it can be concluded that the loading of the particles was successful.

It is also clear that the incorporation of CXB in the two different carrier materials improves the thermal stability of the substance. This behavior has been observed for other substances incorporated in mesoporous structures and might find its explanation in the physics behind the Kelvin equation as discussed by Zhang et al. (2014) [13].

### 2.2. Release and Dissolution of CXB

The dissolution profiles of free and formulated CXB are presented in Figure 5, where supersaturated states of CXB are evident for the formulated samples. A rapid dissolution of CXB was observed for MMC-CXB and N-CXB as compared to the free drug, with N-CXB showing the most pronounced burst release. The initial burst indicates that the dissolution kinetics of the incorporated CXB is much faster than for the free drug. The more pronounced supersaturation state for CXB formulated with Neusilin is probably due to the larger pore size distribution in the material compared to that in MMC. A larger average pore size enhances the constrictivity to tortuosity ratio and, hence, the effective diffusion coefficient of the drug. These results are consistent with those of a previous study [16].

After reaching a maximum dissolved amount of CXB of 9 and 13.5% after 20 and 10 min for MMC-CXB and N-CXB, respectively, a decrease in the profiles could be observed. This was more pronounced for the N-CXB sample, indicating crystallization of the initially dissolved drug in the solution. For the MMC-CXB sample, recrystallization was not as pronounced as for N-CXB, and a plateau-like region could be seen in the curve almost directly after c_max_ was reached. This indicates a continuous release of CXB at the same rate as the rate of recrystallization of the free CXB in solution.

A summary of the calculated kinetic parameters (t_max_, c_max_ and AUC_0–300_) from the three dissolution profiles is shown in Table 1. N-CXB gives higher c_max_ and lower t_max_ compared to MMC-CXB. N-CXB and MMC-CXB has a higher c_max_ and lower t_max_ compared to pure CXB, which confirms that CXB dissolves both faster and in larger amounts when stabilized in its amorphous state.

Both formulations have comparable AUC_0–300_ values (735.6 and 737.8 mg min L^−1^ for MMC-CXB and N-CXB, respectively). This is due to the slower and more continuous release of CXB from MMC after the initial burst compared to N-CXB formulation. The AUC for the two loaded formulations is about 2.5 times higher than that of pure CXB. From these results, it is clear that loading CXB into MMC and Neusilin in its amorphous state enhances the dissolution kinetics of CXB.

### 2.3. Permeability of CXB across Caco-2 Cells

Figure 6 presents the transfer over the Caco-2 cell membrane of both drug-loaded samples. The cumulative concentration of CXB from MMC is linear over time (R^2^ of 0.999) whereas for N-CXB, it is initially rapid (first 15 min) and then declines. A possible explanation for this might be that the dissolution rate of CXB is initially higher and more rapid when loaded in Neusilin, as can be seen in Figure 5, which has larger pores and a broader pore size distribution than those of MMC. Therefore, Neusilin provides a higher initial flux over the cell membrane, whereas when CXB is formulated with MMC, the release profile allows for a more continuous and less variable transfer across the cell monolayer.

Crystalline CXB (same concentration as the drug-loaded samples) and a CXB solution of 130 µg/mL were also subject to the Caco-2 cell membrane. No detectable signals were observed, and a possible explanation could be that crystalline CXB is not soluble in the cell media and can therefore not transfer over the Caco-2 cell membrane. This indicates that CXB needs an enabling formulation strategy to become efficiently absorbed. Similar results have been observed by Riikonen et al. (2015) [26] and Ventura et al. (2005) [27].

The non-loaded samples did not affect the integrity of the cells during the time of the experiments. However, for the drug-loaded samples, a loss of cell layer integrity was observed at 60 min, possibly affecting the flux over the membrane. Integrity was measured by mannitol apparent permeability for which Papp was <8 × 10^−8^ cm/s for CXB suspension and MMC loaded with CXB but 10-fold higher and above the cutoff value for integrity for Neusilin loaded with CXB (9 × 10^−7^ cm/s) for 60 min. All were <3 × 10^−7^ cm/s at the 30-min time point.

Therefore, the flux was only measured for 30 min for the drug-loaded carriers. The safe use of the carriers as such indicates that high local concentrations of CXB were reached during the permeation measurements, which caused the integrity loss—not the carriers themselves. It is worth mentioning that the Caco-2 model is a very sensitive cell monolayer that does not withstand similar treatment as the full epithelium where the tissue is connected to support cells and a protective mucus lining. That the cell monolayer was not able to withstand 60-min direct interaction with a high concentration of CXB does not indicate that the formulations would affect the cells in vivo.

These results indicate that MMC may be helpful in increasing the bioavailability of CXB and that it is a promising, novel excipient in drug delivery applications for poorly soluble compounds.

## 3. Materials and Methods

### 3.1. Materials

Neusilin US2 was gifted from Fuji Chemicals, Toyama, Japan. MgO (product name: MgO N50) was gifted from Lehmann & Voss & Co, Hamburg, Germany. Crystallin CXB was purchased from 3Way Pharm Inc., Shanghai, China. The purity of celecoxib was determined by HPLC to 93%. Monobasic potassium phosphate and sodium hydroxide were purchased from Sigma-Aldrich (St. Louis, MO, USA). All chemicals were used as received.

### 3.2. Synthesis of MMC

MMC was synthesized as described previously [14,15]. Briefly, MgO and CH_3_OH were mixed in a 1:15 weight/volume ratio under a CO_2_ pressure at 55 °C for four days. The temperature was subsequently lowered to room temperature and the reactor was depressurized. The product was dried at 70 °C and then calcined at 250 °C for 6 h. Calcination is needed for complete decomposition of the organic intermediates formed in the reaction carried out in the pressure reactor [15]. Magnesium carbonate is formed as a result of this decomposition. After calcination, the obtained material was grinded in a Planetary Ball Mill (Restch PM; Restch, Haan, Germany) to reduce the size of the particles. The ground material was thereafter sieved to a particle size of 25–50 μm using two sieves, 25 and 50 μm (Restch PM; Restch, Haan, Germany).

### 3.3. Drug Loading

Two grams of CXB was dissolved in 50 mL ethanol after which four grams of either MMC or Neusilin was added to the solution, and the solvent was evaporated at 75 °C using a rotary evaporator. The theoretical drug-loading degree in the two formulations was 33.3 wt%. The two samples, denoted as MMC-CXB and N-CXB, were stored at 70 °C to avoid adsorption of moisture.

### 3.4. Gas Sorption

N_2_ sorption analysis was carried out at −196 °C using an ASAP 2020 from Micromeritics on unloaded and loaded samples. The samples were degassed at a vacuum lower than 10 μm Hg at 130 °C for 6 h. The specific surface area (SSA) was calculated using the multipoint BET method [25], while the pore size distribution was calculated based on the density functional theory (DFT) method using the model for N_2_ at −196 °C for slit-shaped pores. The total pore volume was obtained from single-point adsorption at a relative pressure P/P_0_ ≈ 1. These calculations were all performed using ASAP 2020 (Micromeritics) software.

### 3.5. TGA

TGA was performed in order to confirm the drug-loading degree in the samples and was carried out on a Mettler Toledo, model TGA/SDTA851e, under airflow in an inert alumina cup. The samples were heated from room temperature to 800 °C at a heating rate of 10 °C min^−1^. Equation (1) was used to calculate the total theoretical weight loss of the filled samples, assuming a 33.3 wt% loading degree in the formulated samples.
(1)$$\mathrm{Total}\;\mathrm{weight}\;\mathrm{loss}=\frac{1}{3}*\mathrm{weight}\;\mathrm{loss}\;\mathrm{CXB}\;\left(\%\right)+\frac{2}{3}*\mathrm{weight}\;\mathrm{loss}\;\mathrm{carrier}\;\left(\%\right)$$

### 3.6. XRD

XRD analysis was performed with a Bruker D8 TwinTwin instrument using Cu–K_α_ radiation (λ = 0.154 nm). Samples were ground and placed on silicon zero background sample holders prior to analysis. The instrument was set to operate at 40 kV and 40 mA. Analyses of the XRD-patterns were performed using the software EVA V2.0 from Bruker.

### 3.7. DSC

DSC was performed on a DSC Q2000 instrument (TA Instruments, New Castle, DE, USA) on MMC and Neusilin before and after incorporation of drug. Crystalline CXB was also studied by DSC. Samples of 2.1–4.3 mg were weighed into 5 mm aluminum pans and sealed. The samples were first cooled to −35 °C and then heated to 200 °C at a heating rate of 10 °C min^−1^. The instrument was calibrated for the melting point and melting enthalpy of indium (156.6 °C and 28.4 J g^−1^, respectively). The heat flow was normalized against the amount of CXB in the samples, and the melting enthalpy for crystalline CXB was obtained by integration of the endothermic peak at around 164 °C. The degree of CXB crystallinity in the samples was calculated by comparing the melting enthalpy for free and loaded CXB in the carrier particles.

### 3.8. Drug Release Measurement

The release of CXB was measured in a USP-2 dissolution bath (Sotax AT7 Smart, Sotax AG, Basel, Switzerland) equipped with 1000 mL vessels (37 °C, 250 rpm). Samples with a total drug content of 40 mg CXB (121 mg of either MMC-CXB or N-CXB) were placed in vessels containing 1000 mL phosphate buffer (pH = 6.8). Aliquots of 3 mL were withdrawn from each vessel at regular intervals and filtered before the drug concentration in the liquid samples was analyzed using UV/visual absorbance spectroscopy at 252 nm (UV-1800, Shimadzu Corporation, Kyoto, Japan). This wavelength was chosen based on a UV/vis spectrum of CXB dissolved in ethanol. Phosphate buffer (pH = 6.8) was used as the blank. All measurements were made in triplicates. When analyzing the dissolution profiles, the area under the release %—time curves (AUC) were calculated according to the Trapezoid method using Excel software (Version 16.46). The c_max_ and t_max_ -values for the different samples were extracted directly from the dissolution profiles.

### 3.9. Permeation Measurement

The permeation study was conducted as in Hubatsch et al. (2007) [28].

Caco-2 cells, obtained from American Type Culture Collection (Manassas, Virginia), were cultured in an atmosphere of 90% air and 10% CO_2_. The cells (passage 95 to 105) were seeded on permeable polycarbonate filter supports (0.45 µm pore size, 12 mm diameter; Transwell Costar, Sigma-Aldrich) at a density of 44,000 cells/cm^2^ in Dulbecco’s modified Eagle’s medium supplemented with 10% fetal calf serum, 1% minimum essential medium nonessential amino acids, penicillin (100 U/mL) and streptomycin (100 µg/mL). Monolayers were used for experiments on day 23 after seeding.

[^14^C]-mannitol was used as a paracellular marker to explore the integrity of the cell monolayers after being exposed to MMC, MMC loaded with CXB, Neusilin and CXB-loaded Neusilin. All carriers were studied at 1 mg/mL. CXB suspension (0.23 mg/mL) was used as control. Since MMC increases the pH of HBSS when dissolving, the pH was adjusted with 5 M HCl, resulting in pH 7.4 when 1 mg/mL of MMC was added. All solutions were pre-warmed to 37 °C, and the cells were washed with Hank’s balanced salt solution (HBSS; pH 7.4) and equilibrated for 15 min prior to the integrity and permeation experiment. The HBSS was removed, and the filters with the cell monolayers were transferred to wells containing 1.2 mL of fresh, pre-warmed HBSS (pH 7.4). The suspensions (0.4 mL) were added to the apical side, and samples were drawn from the basolateral chamber at predefined time intervals for 60 min. For integrity studies, 600 µL was sampled from the basolateral chamber and replaced with fresh HBSS. In drug transport studies, the corresponding sample volume was 100 µL. All measurements were made in triplicate.

A ThermoFinnigan TSQ Quantum Discovery triple−quadrupole (electrospray ionization) coupled to a Waters Acquity UPLC instrument was used for concentration determination of samples. A gradient was used (5% mobile phase B to 95% over 2 min total run) on a Acquity BEH C18 column, with mobile phase A consisting of 0.1% formic acid in water and mobile phase B of 0.1% formic acid in acetonitrile. The flow rate was 0.5 mL/min. Samples of 5 μL were injected, and the CXB was monitored in the positive ionization mode for *m/z* 382.10 > 281.82. [^14^C]-mannitol samples were analyzed in a liquid scintillation counter (1900CA TriCarb; PerkinElmer Life Sciences (Waltham, MA, USA)). The obtained data were used to determine the amount of CXB (µg) that permeated.

## 4. Conclusions

The results presented herein shows that MMC suppresses the crystallization of CXB effectively, which results in improved dissolution kinetics of the API when tested in vitro (phosphate buffer, pH 6.8). These results are in good agreement with earlier studies where MMC has been shown to be able to stabilize several APIs in their amorphous states, resulting in an enhanced dissolution rate of the APIs. The suppressed crystallization of CXB also results in an increase in transfer over a Caco-2 cell membrane mimicking the membrane in the small intestine. The MMC formulation allowed for a linear release over the time period explored (30 min). In contrast, reference material Neusilin US2 produced a more rapid release initially, which thereafter decreased. The latter carrier has a larger pore size and broader pore size distribution, which may contribute to the different release and flux pattern observed. In comparison, the linear and less variable flux of the CXB-loaded MMC indicates that MMC may be helpful in producing an increased bioavailability and continuous release of CXB. The translation of these findings to the in vivo situation and to what extent MMC is contributing to increased absorption and bioavailability are subject to investigations in future work.

## Figures and Tables

**Figure 1 molecules-26-06353-f001:**
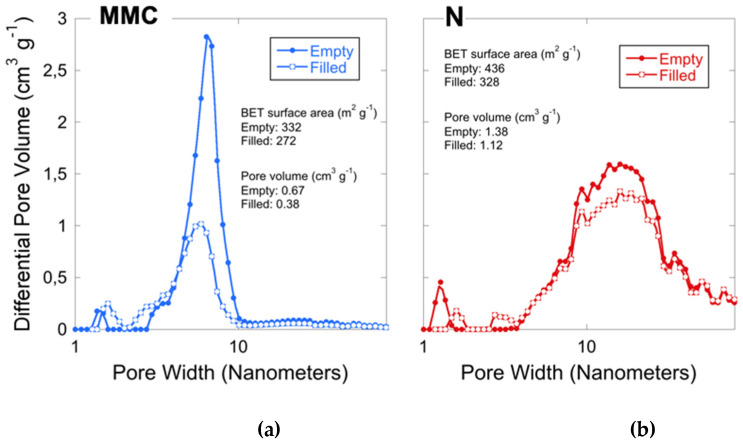
Pore size distribution for empty and loaded (**a**) MMC and (**b**) Neusilin. The respective surface area and pore volume are also displayed in the figures.

**Figure 2 molecules-26-06353-f002:**
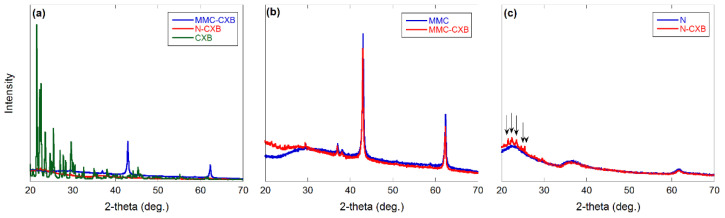
XRD patterns for (**a**) free and loaded CXB and empty and filled (**b**) MMC and (**c**) Neusilin.

**Figure 3 molecules-26-06353-f003:**
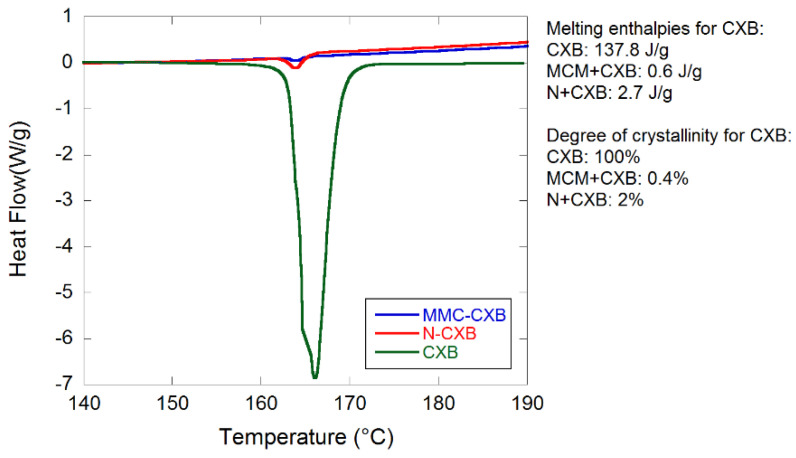
DSC scans of pure and loaded CXB in MMC and Neusilin.

**Figure 4 molecules-26-06353-f004:**
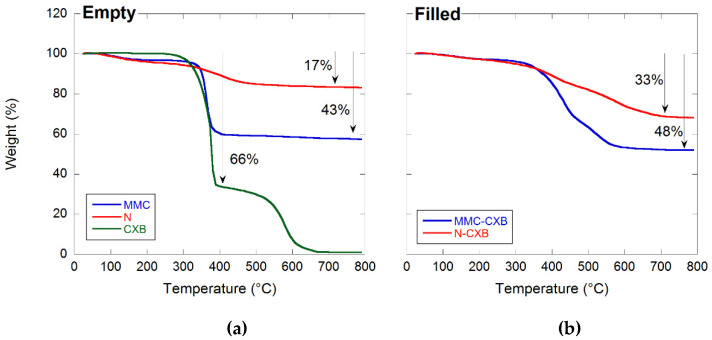
TGA curves for free CXB and the two (**a**) empty and (**b**) filled carrier materials.

**Figure 5 molecules-26-06353-f005:**
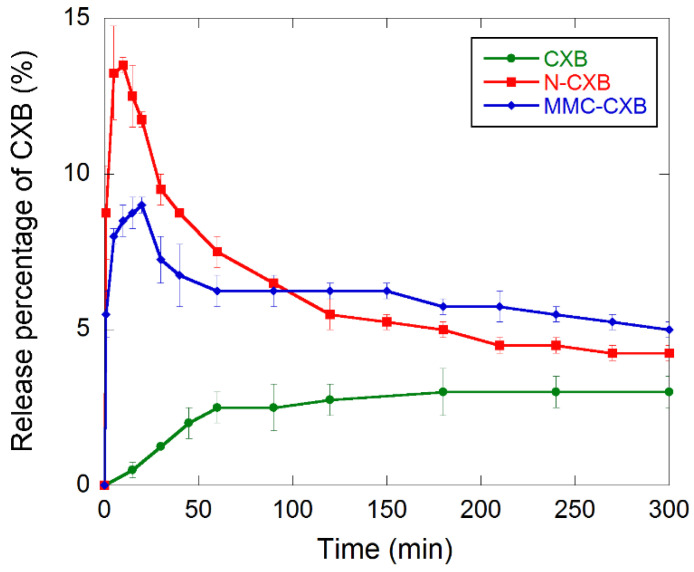
Dissolution profiles for celecoxib in phosphate buffer pH 6.8. Data are presented as mean concentration with error bars representing standard deviations (*n* = 3).

**Figure 6 molecules-26-06353-f006:**
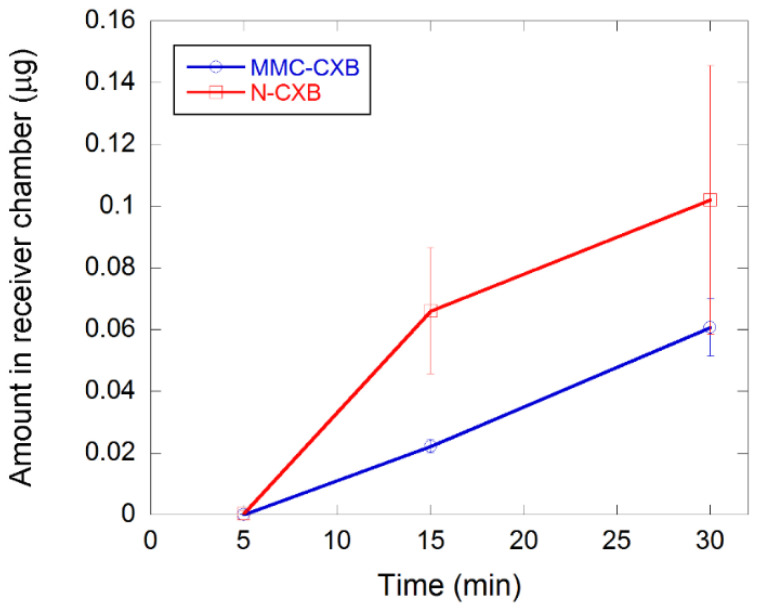
Transport of CXB from MMC and Neusilin across the Caco-2 membrane. The initial CXB concentration in both samples is 300 μg. No detectable signal was observed for crystalline CXB. Data are presented as mean concentration with error bars representing standard deviations (*n* = 3).

**Table 1 molecules-26-06353-t001:** Summary of calculated kinetic parameters using the interval from 0 to 300 min. Abbreviations: t_max_, time to reach maximum concentration; c_max_, maximum concentration, AUC, area under the curve.

	t_max_ (min)	c_max_ (mg L^−1^)	AUC_0–300_ (mg min L^−1^)
Pure CXB	180	1.2 ± 0.1	306.2
MMC-CXB	20	3.6 ± 0.1	735.6
N-CXB	10	5.4 ± 0.1	737.8

## Data Availability

Not applicable.

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
