# Peer review of "Increasing the Transport of Celecoxib over a Simulated Intestine Cell Membrane Model Using Mesoporous Magnesium Carbonate"

_molecules, 2021, doi:10.3390/molecules26216353_

Round 1

Reviewer 1 Report

The work entitled “Increasing the transport of celecoxib in a simulated intestine cell culture model using mesoporous magnesium carbonate” developed and characterized mesoporous magnesium carbonate (MMC) particles for improved drug delivery. The use of the particles was tested for the permeability of the drug celecoxib (CXB) through Caco-2 cells.

Although the study is of potential interest, in my opinion, the results have several issues that should prevent its publication at this stage.

Major comments:

Introduction - It is apparent that MMC has already been studied before, and if CBX permeation is not a problem, it is not clear what the motivation of this work is. Authors should clarify the motivation for carrying out the work in the introduction.

Section 2.2 - The results of figure 5 and their discussion are a mixture of two different concepts that are not clearly distinguished. One is the release of the drug from the particles. The other is the solubility of the drug upon release from the particles. In my opinion, the curve for the release should be a cumulative release, with a monotonic increase over time (similar shape to that obtained for CXB only). If the amount of drug released maintains a soluble form should be a different set of data. The discussion of section 2.2 should be rewritten.

Section 2.3 - In my opinion there are several problems with the permeability results shown in Figure 6. Data for the integrity of the cell barrier must be shown. The permeation time should start counting at 0, not at 5 minutes. It is hard to believe that CXB permeation has not been observed for CXB alone, since its permeation is considered to be fast (see Introduction), and therefore, this would not be a problem. In fact, CXB is a drug that is on the market, so it must have a permeation coefficient that must allow its permeation to happen. Based on the curves obtained, the permeation coefficients of the various forms of CXB must be calculated and compared with literature values, in order to validate the values obtained. In the permeation curves, the deviations from linearity occur after a permeation of about 10% of the amount of starting compound. This is clearly not the case of the experiments on this work. So, one reason for the deviation from linearity is the high error associated to the N-CXB data, so eventually the deviation from linearity is due to experimental error.

Conclusions – Following the description of the article, it seems that the combination of particles plus CXB is harmful to the integrity of the barrier. Therefore, this combination should not be advisable for further use without further improvements.

Minnor comments:

Please check the abbreviations, for instance BET.

Line 308 – “that had been absorbed” -> “that permeated”?

Equation 1, for weight loss implicitly assumes that the weight loss propensity of the CXB is equal to that of the carrier. Please, discuss if this a valid assumption, and its implications?

Author Response

Dear Editor,

We hereby re-submit our manuscript entitled Increasing the transport of celecoxib over a simulated intestine cell membrane model using mesoporous magnesium carbonate (Manuscript ID: molecules-1404656) by Johan Gómez de la Torre, Christel Bergström and Teresa Zardán Gómez de la Torre for publication as an article in Molecules.

We thank the reviewer for the valuable comments. The issues raised by the reviewer have been delt with and included in the revised manuscript where appropriate.

Below you find all questions listed together with our response to each and one of them. Text changes are marked up using the “Track Changes” function in MS Word throughout the revised manuscript.

 Point 1: Introduction - It is apparent that MMC has already been studied before, and if CBX permeation is not a problem, it is not clear what the motivation of this work is. Authors should clarify the motivation for carrying out the work in the introduction.

Answer 1: We thank the reviewer for the comment. We have now clarified the aim of the study. The following changes have been made in the text:

“The objective of this study is to investigate how the previously demonstrated improvement in dissolution kinetics of the poorly soluble model substance celecoxib (CXB) when formulated with MMC affects the flux of CXB in Caco-2 cell model. The Caco-2 cell model is a well characterized intestinal in vitro model that make it possible to evaluate the ability of APIs to cross intestinal barrier. This cell model is widely used in the pharmaceutical industry during drug discovery and development as a predictive tool for the oral absorption of API candidates. [20–23]. A flux increase over the Caco-2 cells could potentially indicate that the bioavailability of CXB can be improved in vivo.”

Point 2: Section 2.2 - The results of figure 5 and their discussion are a mixture of two different concepts that are not clearly distinguished. One is the release of the drug from the particles. The other is the solubility of the drug upon release from the particles. In my opinion, the curve for the release should be a cumulative release, with a monotonic increase over time (similar shape to that obtained for CXB only). If the amount of drug released maintains a soluble form should be a different set of data. The discussion of section 2.2 should be rewritten.

Answer 2: Thank you for this valuable comment. We have rewritten the text in order not to confuse different concepts with each other.We have chosen not to use sink conditions in the release experiments as this enables us to investigate both how the substance is released and simultaneously study any potential supersaturation state. We have included this in the manuscript.

Point 3: Section 2.3 - In my opinion there are several problems with the permeability results shown in Figure 6. Data for the integrity of the cell barrier must be shown. The permeation time should start counting at 0, not at 5 minutes. It is hard to believe that CXB permeation has not been observed for CXB alone, since its permeation is considered to be fast (see Introduction), and therefore, this would not be a problem. In fact, CXB is a drug that is on the market, so it must have a permeation coefficient that must allow its permeation to happen. Based on the curves obtained, the permeation coefficients of the various forms of CXB must be calculated and compared with literature values, in order to validate the values obtained. In the permeation curves, the deviations from linearity occur after a permeation of about 10% of the amount of starting compound. This is clearly not the case of the experiments on this work. So, one reason for the deviation from linearity is the high error associated to the N-CXB data, so eventually the deviation from linearity is due to experimental error.

Answer 3: The carriers here deliver the compound loaded in its amorphous form. The dose given was compared to the same dose of crystalline celecoxib, delivered as a suspension. In this case, no quantifiable permeation was observed within 30 min with a small peak, below limit of quantification, being visible in the 60 min sample. For celecoxib given as solution, the permeability of celecoxib has been reported as 4.6x10-6 cm/s. Papp cannot be determined from the current study as the dissolved C is not known in any of the cases - a prerequisite for the Papp calculation is to know your donor C available for absorption – therefore, instead the flux is reported here. It is likely that the flux is delayed for e.g., the suspension due to it being dictated by the slow dissolution of this poorly soluble drug.The reason for plotting the first value at 5 minutes is because there was no detectable signal at 0 minutes.

Point 4: Conclusions – Following the description of the article, it seems that the combination of particles plus CXB is harmful to the integrity of the barrier. Therefore, this combination should not be advisable for further use without further improvements.

Answer 4: Thank you for the comment. As described in the manuscript, a monolayer of cells is much more sensitive to a suspension and/or particles administered directly to it than would a tissue protected by the mucus barrier be. Therefore, the utility for any of the formulations need to be further evaluated in the next step in animal models with fully mucus protected and a healthy intestine. In fact, we are currently planning to conduct an in vivo study in order to see how these results translate in a “real” situation.

Point 5: Please check the abbreviations, for instance BET.

Answer 5: We have checked all the abbreviations and added their full meaning.

Point 6: Line 308 – “that had been absorbed” -> “that permeated”?

Answer 6: We have changed “that had been absorbed” to “that permeated”

Point 7: Equation 1, for weight loss implicitly assumes that the weight loss propensity of the CXB is equal to that of the carrier. Please, discuss if this a valid assumption, and its implications?

Answer 7: Yes, this is a valid assumption. The wight loss of the pure substance and the empty carriers are established in the experiments (Fig 4a) before the formulations are studied afterwards (Fig 4b).

We hope that our manuscript will be consider for further publication in Molecules after our revision.

With sincere regards,

Teresa Zardán Gómez de la Torre, PhD

Reviewer 2 Report

The main points, which could be improved are:

  1. Please clarify the goal of the study.
  2. Please justify the use of the apparatus 2 in dissolution experiment. The dissolution profiles of CXP suggest that the sink conditions have not been maintained during the study. I would recommend to use apparatus 4 with open loop or apparatus 3.

Author Response

Dear Editor,

We hereby re-submit our manuscript entitled Increasing the transport of celecoxib over a simulated intestine cell membrane model using mesoporous magnesium carbonate (Manuscript ID: molecules-1404656) by Johan Gómez de la Torre, Christel Bergström and Teresa Zardán Gómez de la Torre for publication as an article in Molecules.

We thank the reviewer for the valuable comments. The issues raised by the reviewer have been delt with and included in the revised manuscript where appropriate.

Below you find all questions listed together with our response to each and one of them. Text changes are marked up using the “Track Changes” function in MS Word throughout the revised manuscript.

 Point 1: Please clarify the goal of the study.

Answer 1: We thank the reviewer for the comment. We have now clarified the aim of the study. The following changes have been made in the text:

“The objective of this study is to investigate how the previously demonstrated improvement in dissolution kinetics of the poorly soluble model substance celecoxib (CXB) when formulated with MMC affects the flux of CXB in Caco-2 cell model. The Caco-2 cell model is a well characterized intestinal in vitro model that make it possible to evaluate the ability of APIs to cross intestinal barrier. This cell model is widely used in the pharmaceutical industry during drug discovery and development as a predictive tool for the oral absorption of API candidates. [20–23]. A flux increase over the Caco-2 cells could potentially indicate that the bioavailability of CXB can be improved in vivo.”

Point 2: Please justify the use of the apparatus 2 in dissolution experiment. The dissolution profiles of CXP suggest that the sink conditions have not been maintained during the study. I would recommend to use apparatus 4 with open loop or apparatus 3.

Answer 2: The dissolution test was not performed under sink condition. By using non-sink conditions, we were able to study the super saturation state and crystallization of CXB after release in the dissolution media. This justifies the use of apparatus 2. We have included this in the manuscript.

We hope that our manuscript will be consider for further publication in Molecules after our revision.

With sincere regards,

Teresa Zardán Gómez de la Torre, PhD

Reviewer 3 Report

The article "Increasing the transport of celecoxib in a simulated intestine cell culture model using mesoporous magnesium carbonate," by Johan Gómez de la Torre et al., provides evidence that MMC may be helpful in increasing the bioavailability and obtaining a continuous release of CXB and similar substances, showing some attractive points for readers of Molecules. However, the authors are suggested to improve the manuscript in considering the following issues.

  1. In the Material characterization section, it is suggested to use of scanning electron microscope (SEM) to observe the pore size of MMC materials;
  2. There is no data to support the improvement of CXB solubility after MMC modification. The authors should offer relevant experiments to verify this conclusion;
  3. In line 182-183, "a loss of cell Layer integrity was observed at 60 minutes", the authors should offer relevant data in this manuscript;
  4. The authors should include "n" values for all groups in Figure 6, as well as the number of replicates per experimentï¼›
  5. Intestinal microenvironment mainly includes acid-base balance, microbial balance. In this paper, the author considers the intestinal pH influence of MMC-CXB. However, the authors did not consider the effect of intestinal microbes on MMC-CXB. It is preferable to have a separate Discussion section, which compares the effects of Neusilin US2 and MMC materials on CXB and the effects of intestinal microbes on MMC-CXBï¼›
  6. The manuscript contains many grammatical errors, and would benefit from careful editing for the use of English language.

Author Response

Dear Editor,

We hereby re-submit our manuscript entitled Increasing the transport of celecoxib over a simulated intestine cell membrane model using mesoporous magnesium carbonate (Manuscript ID: molecules-1404656) by Johan Gómez de la Torre, Christel Bergström and Teresa Zardán Gómez de la Torre for publication as an article in Molecules.

We thank the reviewer for the valuable comments. The issues raised by the reviewer have been delt with and included in the revised manuscript where appropriate.

Below you find all questions listed together with our response to each and one of them. Text changes are marked up using the “Track Changes” function in MS Word throughout the revised manuscript.

 Point 1: In the Material characterization section, it is suggested to use of scanning electron microscope (SEM) to observe the pore size of MMC materials.

Answer 1: MMC is a non-conducting material, hence, SEM operation at such voltage needed to resolve pores in current the size range is not feasible due to charging of the sample. Sputtering MMC with a conducting material before SEM analysis is also not feasible approach as this would potentially cover the pores. Furthermore, SEM is mainly a method used to study surfaces and cannot give a full picture of the geometric nature of pores.

Point 2: There is no data to support the improvement of CXB solubility after MMC modification. The authors should offer relevant experiments to verify this conclusion.

Answer 2: We thank the reviewer for the valuable comment. The reviewer is right that we can only conclude that the dissolution rate is improved after MMC modification. We have therefor made changes throughout the text.

Point 3: In line 182-183, "a loss of cell Layer integrity was observed at 60 minutes", the authors should offer relevant data in this manuscript

Answer 3: Integrity was measured by mannitol apparent permeability for which Papp was <8x10-8 cm/s for celecoxib suspension and MMC loaded with celecoxib but 10-fold higher and above the cutoff value for integrity for Neusilin loaded with celecoxib (9x10-7 cm/s) for 60 min. All were <3x10-7 cm/s at 30-minute time point. This information is now added to the manuscript.

Point 4: The authors should include "n" values for all groups in Figure 6, as well as the number of replicates per experiment

Answer 4: We have provided the number of replicates (n value) for all groups in connection to Figure 6.

Point 5: Intestinal microenvironment mainly includes acid-base balance, microbial balance. In this paper, the author considers the intestinal pH influence of MMC-CXB. However, the authors did not consider the effect of intestinal microbes on MMC-CXB. It is preferable to have a separate Discussion section, which compares the effects of Neusilin US2 and MMC materials on CXB and the effects of intestinal microbes on MMC-CXB

Answer 5: Thank you for the comment. The objective of this study was to investigate how the improvement in dissolution kinetics of the model substance celecoxib (CXB) when formulated with MMC translates into an increased flux in a caco-2 model. This could potentially indicate that the bioavailability of CXB can be improved in vivo. Even though we believe that an analysis of the effect of intestinal microbes on MMC and Neusilin is of great interest, it falls outside of the scope of this study.

Point 6: The manuscript contains many grammatical errors, and would benefit from careful editing for the use of English language.

Answer 6: Thank you for this valuable comment. We have carefully read through the entire manuscript and made appropriate changes.

We hope that our manuscript will be consider for further publication in Molecules after our revision.

With sincere regards,

Teresa Zardán Gómez de la Torre, PhD

Round 2

Reviewer 1 Report

I have no further comments.

Reviewer 3 Report

The points raised were addressed and the manuscript can be suitable for publication in the present form.

This manuscript is a resubmission of an earlier submission. The following is a list of the peer review reports and author responses from that submission.

Round 1

Reviewer 1 Report

The manuscript describes the use of MMC as a means to transport the drug celecoxib by in vitro assays on a Caco-2 cell membrane that simulates the membrane of the small intestine, compared to the use of Neusilin US2.

In my opinion the manuscript should be published as it is a good contribution to the development of potential drug carriers. However, there are some aspects that must be clarified. For instance:

The section number 3.3 corresponding to “Permeability of CXB across Caco-2 cells”, should be changed to the section number 2.3 (line 163).

Also, Reference 13, should be properly spelled for the first author: Zhang, instead of Peng (line 104).

Reviewer 2 Report

From the perspective of the content of the title and abstract, I am very interested in this manuscript and think it is innovative and a good entry point. But the format of this manuscript obviously does not conform to the format of the paper. I think the source of the material, the experimental method, and the calculation formula of the data are very important parts and need to be separated separately. The current description in this part is very vague and mixed with the results part. Therefore, I think it cannot be published until it is revised to a standard format.

Reviewer 3 Report

The paper titled "Increasing the transport of celecoxib in a simulated intestine model using mesoporous magnesium carbonate" highlights the potential of mesoporous magnesium carbonate (MMC) in enhancing the solubility and dissolution profile of celecoxib. The reported approach lacks novelty since MMC has shown the same effect with celecoxib and other poorly soluble drugs in published work.  

Reviewer 4 Report

The authors developed a mesoporous material for celecoxib transportation in simulated intestine which looks very promising. However, I would like to ask the authors to explain some of my questions as below:

  1. Line 211-212: should illustrate sieving procedure in detail. Did you use double-sieving to determine the upper (50 µm) and lower (25 µm) limit of particle size or measured particle size after sieving;
  2. About cell line experiments, could you please provide a reference to explain why you choose to use 60 min as the standard of cytotoxicity. As far as I known, cytotoxicity experiments are usually conducted for a duration of 12 h or 24 h with a surviving rate no less than 90% (Wolfe & Liu, 2007);
  3. The results of Transport of CXB from Neusilin across the Caco-2 membrane in Figure 6 have fairly large standard deviation which, in my opinion, is enough to influence the analysis. Therefore, I think this part of experiments should be repeat if possible.
  4. Did you develop the methods all on your own? Otherwise, please add references to methods;
  5. It looks like you obtained your analysis and conclusion with little support from others. Please add references to results and discussion section which can support your results or hypotheses.

Reviewer 5 Report

Gomez de la Torre has described in the manuscript the utility of MMC as potential to increase intestinal solubility of celecoxib. However, there is no novelty with the work, the group has already published the same setup of experiments with the same compound. Additionally, the title is misleading, there is no additional investigation regarding simulated intestinal fluid apart from the Caco2 experiment. In this regard, as well there is no description of the experiment with Caco2 in the manuscript. Would be interesting to get the formulation tested in vivo and in biorelevant media to show the novelty of the formulation. Also, the discussion section is a bit bland needs more improvements.